# Differences in Mutational Profile between Follicular Thyroid Carcinoma and Follicular Thyroid Adenoma Identified Using Next Generation Sequencing

**DOI:** 10.3390/ijms20133126

**Published:** 2019-06-26

**Authors:** Martyna Borowczyk, Ewelina Szczepanek-Parulska, Szymon Dębicki, Bartłomiej Budny, Frederik A. Verburg, Dorota Filipowicz, Barbara Więckowska, Małgorzata Janicka-Jedyńska, Lidia Gil, Katarzyna Ziemnicka, Marek Ruchała

**Affiliations:** 1Department of Endocrinology, Metabolism and Internal Diseases, Poznan University of Medical Sciences, 60-355 Poznań, Poland; 2Department of Nuclear Medicine, University Hospital Marburg, 35043 Marburg, Germany; 3Department of Computer Science and Statistics, Poznan University of Medical Sciences, 60-806 Poznań, Poland; 4Department of Clinical Pathology, Poznan University of Medical Sciences, 60-355 Poznań, Poland; 5Department of Hematology and Bone Marrow Transplantation, Poznan University of Medical Sciences, 60-569 Poznań, Poland

**Keywords:** follicular thyroid cancer, follicular thyroid adenoma, next-generation sequencing, whole-genome studies, genetics

## Abstract

We aimed to identify differences in mutational status between follicular thyroid adenoma (FTA) and follicular thyroid cancer (FTC). The study included 35 patients with FTA and 35 with FTC. DNA was extracted from formalin-fixed paraffin-embedded (FFPE) samples from thyroidectomy. Next-generation sequencing (NGS) was performed with the 50-gene Ion AmpliSeq Cancer Hotspot Panel v2. Potentially pathogenic mutations were found in 14 (40%) FTA and 24 (69%) FTC patients (OR (95%CI) = 3.27 (1.22−8.75)). The number of mutations was higher in patients with FTC than FTA (*p*-value = 0.03). *SMAD4* and *STK11* mutations were present only in patients with FTA, while defects in *FBXW7*, *JAK3*, *KIT*, *NRAS*, *PIK3CA*, *SMARCB1*, and *TP53* were detected exclusively in FTC patients. *TP53* mutations increased the risk of FTC; OR (95%CI) = 29.24 (1.64–522.00); *p*-value = 0.001. *FLT3*-positivity was higher in FTC than in the FTA group (51.4% vs. 28.6%; *p*-value = 0.051). The presence of *FLT3* and *TP53* with no *RET* mutations increased FTC detectability by 17.1%, whereas the absence of *FLT3* and *TP53* with a presence of *RET* mutations increased FTA detectability by 5.7%. *TP53* and *FLT3* are candidate markers for detecting malignancy in follicular lesions. The best model to predict FTA and FTC may consist of *FLT3*, *TP53*, and *RET* mutations considered together.

## 1. Introduction

The differentiation between follicular thyroid adenoma (FTA) and follicular thyroid carcinoma (FTC) in preoperative diagnostics is not reliable when based on imaging or biopsy [1]. Therefore, in recent years, many studies have attempted to identify additional factors for the discrimination of these two related pathological entities [2], as well as to further clarify the extent of the relationship in their pathogenesis [3].

A main area on which research has focused is the identification of genetic alterations that may be present in FTC, but not in FTA. Molecular markers are increasingly used as pre-surgical diagnostic tools in the management of indeterminate thyroid nodules [4]. However, the panel of genes proposed to date has not demonstrated conclusive diagnostic results [5].

While in papillary thyroid cancer (PTC), next-generation sequencing (NGS) has yielded promising results [6,7], this technique has, thus far, rarely been applied to FTC [8]. The majority of studies in FTC have tested only well-recognized thyroid cancer-related genes using, e.g., a 7-gene [9], 24-gene panel, or standard PCR for single mutations [10]. Furthermore, for FTA, even fewer data about the genetic alterations have been published [11]. To the best of our knowledge, authors comparing the genetic background of FTA and FTC have so far rarely applied NGS [12], which enables detection of many genetic changes simultaneously [13] and the analysis of the complex genetic interplay of distinct mutations [14].

Recently, in oncological analyses, novel extensive panels have been successfully applied in a variety of solid cancers [15]. These have been used to identify point mutations, single nucleotide variants, polymorphisms, and rearrangements, which may be associated with the genesis of malignant tumors [16]. However, the genetic background of FTA and FTC has not yet been compared using this method.

The aim of this study was to compare FTA and FTC using the wide oncological molecular panel and to identify differences in mutational profile, which may aid in the preoperative differentiation between these two related thyroidal pathologies, as well as in an understanding of their pathways of origin.

## 2. Results

The molecular analyses allowed for the identification of at least one possibly pathogenic mutation in 14 out of 35 (40.0%) patients diagnosed with FTA and 24 (68.6%) out of 35 patients diagnosed with FTC (Table 1). The number of detected mutations was significantly higher in patients with FTC in comparison with those diagnosed with FTA (*p*-value = 0.03). Identification of any mutation from the panel indicated that the patient was 3.27-times more likely to be diagnosed with FTC (OR (95%CI) = 3.27 (1.22–8.75)).

The median frequency and the depth of coverage of all detected variants in the whole cohort were 59.9% and 537 x, respectively, and ranged from 10.8% to 100% and from 260 to 11138 x. Twenty-six out of the mutation-positive patients (68%) had allele burden of over 51% and were considered homozygous for the mutation.

The majority of the mutations occurred with the same frequency in FTA and FTC. *SMAD4* and *STK11* mutations were present only in patients diagnosed with FTA. Defects in *FBXW7*, *JAK3*, *KIT*, *NRAS*, *PIK3CA*, *SMARCB1*, and *TP53* were detected exclusively in FTC patients. However, only *TP53* mutations indicated that patients were significantly more likely to be diagnosed with FTC (OR (95%CI) = 29.24 (1.64−522.00), *p*-value = 0.001).

The most common mutation in FTA was in *RET*, followed by *FLT3* (31.4% and 28.6% of all FTA) whereas, conversely, the most common FTC mutation was in *FLT3*, followed by *RET* (51.4% and 31.4% of all FTC). The increased *FLT3*-positivity in FTC was found to be significant (*p*-value = 0.051). Figure 1 shows a list of mutations common to and exclusive for FTA and FTC and the co-occurrence of the mutations in different patients.

The variety of the mutations tended to occur with higher frequency in FTC in comparison with FTA although this was not significant.

Multivariate logistic regression analysis showed that *FLT3*, *TP53*, and *RET* mutations should be jointly considered to improve the prediction of FTA and FTC diagnosis (Table 2 and Table 3). For this model, the area under curve (AUC) in ROC curve was significantly higher for the prognosis of FTA and FTC than in a case of single mutation model (AUC (Model 1) = 0.64, AUC (Model 2) = 0.76, *p*-value = 0.016). These results were also confirmed by IDI (0.11, *p*-value = 0.003), as well as categorical net reclassification improvement (categorical NRI) (0.23, *p*-value = 0.007).

The prevalence of *FLT3* and TP53 with no *RET* mutations increased FTC detectability by 17.1% (NRI (FTC) = 0.171), whereas the absence of *FLT3* and *TP53* with a presence of *RET* mutations increased FTA detectability by 5.7% (NRI (FTA) = 0.057). Assignment of patients to FTC and FTA groups after reclassification due to the logistic regression model based on three indicated mutations is also shown in Figure 2.

Ten patients (28.6%) were correctly classified in the FTC group based only on the *TP53* mutation, whereas 16 (45.7%) patients were correctly classified based on *TP53*, *FLT3*, and *RET*. Similarly, the number of patients correctly classified as belonging to the FTA group increased by 8.6%. based on these three indicated mutations.

The characteristics of patients diagnosed with follicular thyroid cancer according to their mutational status is presented in the Table 4. The full list of mutations, which occurred in at least three FTA and/or FTC samples, is presented in Appendix A.

## 3. Discussion

Our study provides a new perspective in the debate on genetics of follicular lesions. Whether FTA and FTC constitute distinct entities or a continuum is still debatable [17,18,19,20]. The similarity of their genetic background found here may suggest the latter. The majority of mutations occur with similar frequency in FTA and FTC. However, the number of possible pathogenic mutations is higher in FTC than FTA. Also, the variety and number of the mutations found exclusively in FTC tend to be higher than in FTA. This demonstrates a more complex FTC genetics than FTA and may reflect a natural history of FTC with the accumulation of new genetic changes.

The most common finding in FTC was *FLT3* mutation. A mutation in class III receptor tyrosine kinase that is involved in the regulation of apoptosis, proliferation, and differentiation of hematopoietic cells may result in constitutive autophosphorylation of the immature form of the FLT3 receptor, resulting in strong factor-independent activation of STAT5 [21], which was previously described in acute myeloid leukemia (AML) [22,23]. However, our study demonstrates that this may be similar to FTC. We demonstrated that the *FLT3* mutation, most common in FTC, may be an interesting malignancy marker candidate as it was nearly twice as common in FTC than in FTA. As pre- and postsurgical diagnosis of FTC remains a challenge [24], *FLT3*-positivity as a possible marker of malignancy should be carefully evaluated in future prospective studies.

According to our results, *TP53* may be another candidate malignancy marker, as this was the second most common mutation in FTC samples not detected in any FTA specimen. The presence of *TP53* mutation indicated the patient was 29-times more prone to have FTC. The role of *TP53* has been confirmed in previous studies [25]. *TP53* damage in a differentiated carcinoma is likely a key factor of dedifferentiation and major chromosomal instability [26,27]. *TP53* alterations causing inactivation of apoptosis and cell-cycle progression were also reported to be the most frequent (54.4%) in anaplastic thyroid carcinomas (ATCs) [28]. The wide confidence interval for the odds ratio does not allow us to precisely determine how many times higher the risk for FTC diagnosis will be when this mutation occurs, but it does not undermine the statistical significance of the relationship. The significant influence of the *TP53* mutation on the occurrence of FTC was also confirmed by building a logistic regression model and determining the ROC curve (Table 3), where we can also see that the area under the ROC curve with the confidence interval (AUC (95% CI) = 0.64 (0.57−0.72)) is above 0.5, which increases the reliability of the findings.

*RAS* mutations are most common in FTA and FTC, where they are considered to be the second most common genetic alterations [29]. In our study, *RAS* was the third most commonly mutated gene. The RAS pathway is considered to be an indicator of follicular-derived thyroid lesions [30]. However, despite their high prevalence in FTA and FTC, their role as malignancy markers remains unclear [31]. The importance may lie in a type of *RAS* mutation. In our study, *HRAS* occurred both in FTA and FTC. However, *NRAS* was more prevalent in FTC. *NRAS* has previously been reported to be more frequent in carcinomas (24%) than adenomas (14%) [32], and significantly associated with mortality. There is a possibility that the *NRAS* mutation may indicate a higher malignant potential than other forms of *RAS* [33]. However, the presence of the *NRAS* mutation in only one patient sample in our study reduces the generalizability of this finding and requires further research. Moreover, the exclusive occurrence of an *NRAS* mutation in FTC, shown also in other studies [33], indicates that the diagnosis of adenoma alongside the presence of this mutation should be made cautiously. As opposed to previous findings [34,35], *KRAS* was not found in either FTA or FTC. The presence of *HRAS* mutation with the same frequency in both benign and malignant lesions shown in our study may suggest its role in early tumorigenesis [36], and suggests a common genetic background. Our study raises the possibility that FTAs with *RAS* mutations have an inherent malignant potential and suggests caution. However, the clinical significance of this finding should be further investigated in more patients and over a longer follow-up period. The results of our study should not drive the decision for surgery, although they may highlight a group of higher-risk patients that should be more vigilantly monitored, as *RAS*-positive FTAs may have a higher risk of lesion malignant transformation.

An interesting finding is the presence of *SMAD4* and *APC* mutations in FTA, previously reported in ATC [28]. Surprisingly, *RET* mutations, occurring mostly in medullary thyroid carcinoma, were one of the most common genetic events in both FTAs and FTCs. As the mutation was previously found in both benign and malignant thyroid lesions [37], mainly advanced and poorly differentiated thyroid cancers, this may indicate a group of patients who should be subjected to special surveillance. Thus, the mutational profile may determine not only a choice of targeted treatment, but also the frequency of the follow-up visits.

The mutation in the *MET* gene described in both FTA and FTC may potentially be a druggable target. This demonstrates a role for NGS in treatment personalization, to determine treatment based on the mutational profile of the tumor and to choose a drug that would potentially target the detected cancer pathway.

A logistic regression model showed that the most accurate predictive model for FTC and FTA is based on the combination of *FLT3*, *TP53*, and *RET* mutations. The prevalence of *FLT3* and *TP53* with no *RET* mutations increased FTC risk by 17.1%, whereas the presence of *RET* mutations alone indicated an increased FTA detectability by 5.7%. Any of the described factors, apart from the mutational status, do not differ when comparing FTA and FTC—which makes the mutational status the only factor differentiating FTA and FTC in the multivariate regression analysis. When compared to different features of FTC, which is shown in Table 4, no significant clinico-pathological differences in a group of patients divided according to their mutational status was found. Again, the mutational status is the only factor to differ between groups and to be included in multivariate analysis. The prediction of FTA and FTC is, therefore, based on complex genetic interplay of the mutations and reflects heterogenic genetic background of follicular lesions.

Although our study group was relatively small, the research focused on detailed analysis of NGS data of FTA and FTC from homogenous population. This is the first study to compare genetic background in FTA and in FTC from the same population with NGS using a panel that comprises not only thyroid cancer-specific mutations [38], but an extended panel of 50 different types of cancer-related genes. Moreover, given a parameter value difference of 25%, a standard deviation of 30%, an alpha error level of 0.05, and a beta error level of 50%, both FTC and FTA groups should consist of at least 16 patients to obtain statistically significant conclusions. Another limitation was the absence of paired normal tissue samples. However, recent recommendations of the Association for Molecular Pathology and College of American Pathologists (AMPCAP) established that a 30 x medium coverage might be adequate for germline testing [39]. In the absence of paired normal tissue samples, a much higher coverage is necessary for somatic testing, and for targeted panels, the coverage recommended by AMPCAP to reliably report a somatic variant is at least 250 reads depth in the clinical setting [39]. Therefore, such a cut-off was established, although this decreased the number of mutations found.

Further studies are needed to understand the relationships between the presence of these genetic mutations, their impact on the development of thyroid tumors, and their clinical application. Follow-up studies in a larger cohort of patients with a longer follow-up period, including collection of data on radioiodine refractoriness and overall survival rate, are required to fully understand the significance of these mutations in thyroid follicular lesions, as well as their potential as new diagnostic and therapeutic targets.

FTA and FTC share many common mutations. However, the number of mutations is significantly higher in FTC as compared with FTA. This indicates that FTA and FTC may share common genetic background that may be more complex in the case of FTC. *TP53* and *FLT3* may be candidate markers for detecting malignancy in follicular lesions. However, the most accurate predictive model for FTA and FTC may comprise a combination of *FLT3*, *TP53*, and *RET* mutations. Further understanding of the importance of *FLT3*, *HRAS*, *RET*, and other mutations found in FTA is required to determine whether more careful follow-up or more aggressive treatment is necessary. Identification of new genetic factors involved in FTC pathogenesis could improve understanding of carcinogenesis and enable the development of new targeted drugs.

## 4. Materials and Methods

### 4.1. Patients’ Characteristics and Clinicopathological Analysis

We retrospectively analyzed 70 randomly selected patients at the single tertiary care department of endocrinology of our university hospital, diagnosed with follicular lesions (35 with FTA and 35 with FTC), according to the World Health Organization [40]. Both groups were adjusted for age and gender.

The study was approved by the Bioethical Committee of Poznan University of Medical Sciences (an approval no. 1061/15 from January 2015) and was conducted in accordance with the Declaration of Helsinki. Additional informed consent was not required due to the retrospective nature of our analysis and the use of stored materials.

Surgically obtained formalin-fixed paraffin-embedded (FFPE) samples from total or subtotal thyroidectomy and detailed clinical annotates were subjected to further analysis. The group consisted of 58 women and 12 men with a median age at diagnosis of 55 years (range 27 to 82). Patients enrolled in the study were Caucasians not suffering from any other endocrine disorders or cancer and were not receiving any treatment at the time of diagnosis. The analysis covered data collected 2008–2018. Patient characteristics are presented in Table 5.

The diagnosis of FTC was confirmed by histopathological reexamination of the specimen following thyroidectomy performed by qualified pathologists. For each patient, we recorded the age at diagnosis, gender, tumor size, multifocality, extra-thyroidal extension, the presence of histopathological signs of chronic lymphocytic thyroiditis, histopathological staging (pTNM) according to the 8th tumor-node-metastasis (TNM) classification [41], and radioiodine refractoriness.

Tumors were regarded as multifocal when two or more foci were found. In the case of multifocality, the size of the tumor was noted as the size of the largest focus. All samples were anonymized.

For the present study, FTC was defined as radioactive iodine (RAI) refractory if a cumulative activity of 22.2 GBq/600 mCi of RAI did not result in achieving complete remission.

### 4.2. Genomic DNA Extraction

A qualified pathologist indicated areas of interest from 70 FFPE specimens, and the unstained slides were manually microdissected. Genomic DNA was extracted using a QIAamp DNA FFPE Tissue Kit (Qiagen, Valencia, CA, USA), following the manufacturer’s instructions. Genomic DNA was quantified using a fluorometer Qubit platform (Invitrogen, Carlsbad, CA, USA), and the DNA quality and integrity were tested.

### 4.3. Next Generation Sequencing

The 50-gene Ion AmpliSeq Cancer Hotspot Panel v2 (CHPv2) (Thermo Fisher Scientific, Carlsbad, CA, USA) was used with the IonTorrent^TM^ Personal Genome Machine platform (Life Technologies, Foster City, CA, USA) in all experiments. The panel enables the amplification of 207 amplicons covering approximately 2800 mutations deposited in the COSMIC database from 50 oncogenes and tumor-suppressor genes commonly mutated in human cancers (*ABL1*, *AKT1*, *ALK*, *APC*, *ATM*, *BRAF*, *CDH1*, *CDKN2A*, *CSF1R*, *CTNNB1*, *EGFR*, *ERBB2*, *ERBB4*, *EZH2*, *FBXW7*, *FGFR1*, *FGFR2*, *FGFR3*, *FLT3*, *GNA11*, *GNAS*, *GNAQ*, *HNF1A*, *HRAS*, *IDH1*, *IDH2*, *JAK2*, *JAK3*, *KDR*, *KIT*, *KRAS*, *MET*, *MLH1*, *MPL*, *NOTCH1*, *NPM1*, *NRAS*, *PDGFRA*, *PIK3CA*, *PTEN*, *PTPN11*, *RB1*, *RET*, *SMAD4*, *SMARCB1*, *SMO*, *SRC*, *STK11*, *TP53*, and *VHL*). The Ion AmpliSeq Library Kit, version 2.0 (Life Technologies) was used to amplify 10 ng of DNA according to the manufacturer’s instructions. Sequencing beads were templated and enriched using the Hi-Q Template OT2 200 Kit. The libraries were barcoded with Ion Xpress Barcode Adaptors Kit (Thermo Fisher Scientific, Carlsbad, CA, USA), clonally amplified by emulsion PCR in vitro on the Ion PGM Template OneTouch 2 system (Thermo Fisher Scientific, Carlsbad, CA, USA). Ionsphere particles with DNA were isolated and sequenced on an Ion 318v2 Chip using the Hi-Q Sequencing Kit (Thermo Fisher Scientific, Carlsbad, CA, USA) according to the manufacturer’s protocols.

### 4.4. Mutation Analysis

The obtained data from genomic experiments were analyzed using dedicated software. Signal processing, mapping, and quality control were performed with Torrent Suite Software, v.5.2 (Life Technologies). The sequence variants were called, and data were analyzed using the Ion Reporter with the AmpliSeq CHPv2 single-sample workflow and default settings. The Variant Caller plugin included in the Torrent Suite Software (v.3.6; Thermo Fisher Scientific, Carlsbad, CA, USA) and the MutationTaster2 algorithm were used to identify variations in target regions. Variants were categorized according to whether they comprised a nonsynonymous or frameshift mutation or stop codon in the exonic region. Each of the identified genetic variations was coded according to “plus strand” of the Human Genome assembly hg19. The limit of detection was a 5% mutational allelic frequency at 250 × coverage depth for each tested region.

To analyze a putative function of mutations as driver mutations, we employed four separate programs: SIFT [42], Polyphen-2 [43], and MutationTaster2 [44], as well as FATHMM (functional analysis through hidden Markov models (v2.3), which resulted in an index, calculated with a high- throughput web-server, able to predict the functional consequences of both coding variants, i.e., non-synonymous single nucleotide variants (nsSNVs), and non-coding variants to distinguish between cancer-promoting/driver mutations and other germline polymorphisms [45].

We checked for the presence of particular mutations and their previous reports in the catalogue of somatic mutations in cancer (COSMIC), the dbSNP, 1000 Genome Project, ClinVar, and ExAC databases.

### 4.5. Statistical Analysis

Parameters were recorded and entered into a dedicated database. Descriptive analysis was used to summarize the collected data. To determine the normality of continuous variables, data were tested by the D’Agostino and Pearson omnibus normality test. Variables that were found to be normally distributed were expressed as mean values. Data that were found to be distributed differently were expressed as median and minimum–maximum values.

To compare differences between the groups for categorical variables, the chi-square test was used if the Cochrane assumptions were met, or the Fisher’s exact test otherwise. Interval data were compared with the use of the Mann–Whitney U test as the data did not follow a normal distribution. Odds ratio (OR) and the 95% confidence interval (95% CI) were calculated using the group of patients diagnosed with FTAs as the reference population.

Multivariate logistic regression analysis included mutations that appeared more than twice. A backward stepwise logistic regression indicated mutations that should be considered jointly as predictors of FTC and FTA-Model 2. Firth’s method for eliminating the well-known small sample bias in maximum likelihood estimation was applied to calculate logistic regression coefficients. These were then compared with the predictive accuracy of the most significant single mutation alone (Model 1). Using the regression model, patients diagnosed with FTA were treated as a reference group, whereas high-risk values indicated a highly probable occurrence of FTC. The difference was calculated with the ROC curve (DeLong’s method), the integrated discrimination improvement (IDI), and the categorical net reclassification improvement (categorical NRI) [46], for which the patient was classified to the FTC group if the risk exceeded 70% and to the FTA group if it was lower than 30%. No decision taken if the risk was within 30–70%.

A *p*-value of less than 0.05 was regarded as significant. Statistical analyses were performed with StatSoft Statistica v13.0 (stepwise logistic regression), LogXact v11.1.0 (logistic regression coefficients based on penalized maximum likelihood estimation for bias correction-Firth’s method), and PQStat v1.6.8 software (ROC comparison, NRI, IDI, and one-dimensional analysis).

## Figures and Tables

**Figure 1 ijms-20-03126-f001:**
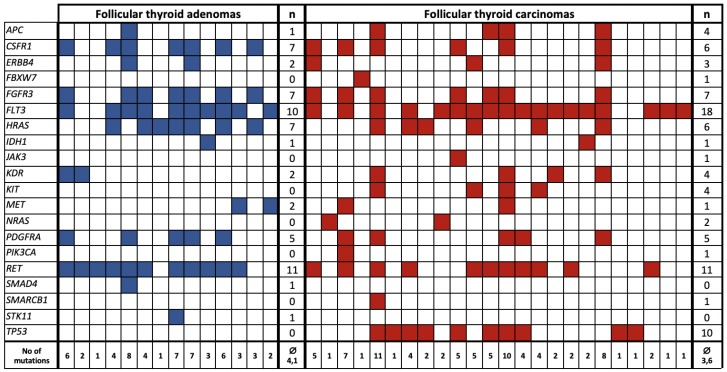
Genetic landscape of follicular thyroid adenomas (*n* = 14/35)—marked in blue—and follicular thyroid carcinomas (*n* = 24/35)—marked in red. Columns of the table represent single patients, rows—type of mutation.

**Figure 2 ijms-20-03126-f002:**
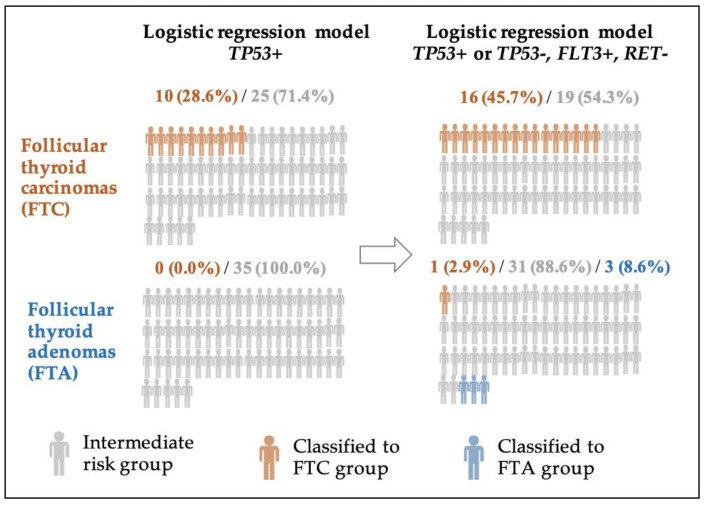
Reclassification of individuals predicted to follicular thyroid cancer (FTC) group and to follicular thyroid adenoma (FTA) group according to logistic regression model based on mutational status.

**Table 1 ijms-20-03126-t001:** Patients’ mutational status according to histopathological diagnosis.

Characteristics	Follicular Thyroid Adenomas *n* = 35	Follicular Thyroid Carcinomas *n* = 35	*p*-Value	OR (95%CI)
**Presence of any mutation, *n* (%)**	**14 (40.0%)**	**24 (68.6%)**	**0.030**	**3.27 (1.22–8.75)**
Presence of any mutation found in COSMIC database, *n* (%)	8 (22.9%)	9 (25.7%)	0.780	1.17 (0.39–3.49)
Number of mutated genes, *n* (%)	13 (37.1%)	18 (51.4%)	0.229	1.79 (0.69–4.65)
Mutations found in both groups of patients				
*APC*, *n* (%)	1 (2.9%)	4 (11.4%)	0.356	4.39 (0.46–41.40)
*CSFR1*, *n* (%)	7 (20.0%)	6 (17.1%)	0.759	0.83 (0.25–2.77)
*ERBB4*, *n* (%)	2 (5.7%)	3 (8.6%)	1.000	1.55 (0.24–9.88)
*FGFR3*, *n* (%)	7 (20.0%)	7 (20.0%)	1.000	1.00 (0.31–3.23)
*FLT3*, *n* (%)	10 (28.6%)	18 (51.4%)	0.051	2.65 (0.99–7.11)
*HRAS*, *n* (%)	7 (20.0%)	6 (17.1%)	0.759	0.83 (0.25–2.77)
*IDH1*, *n* (%)	1 (2.9%)	1 (2.9%)	1.000	1.00 (0.06–16.65)
*KDR*, *n* (%)	2 (5.7%)	4 (11.4%)	0.673	2.13 (0.36–12.46)
*MET*, *n* (%)	2 (5.7%)	2 (5.7%)	1.000	1.00 (1.13–7.53)
*PDGFRA*, *n* (%)	5 (14.3%)	5 (14.3%)	1.000	1.00 (0.26–3.81)
*RET*, *n* (%)	11 (31.4%)	11 (31.4%)	1.000	1.00 (0.36–2.74)
*COSMIC* mutations found in both groups of patients				
*HRAS* (COSM249860), *n* (%)	7 (20.0%)	5 (14.3%)	0.526	0.67 (0.19–2.35)
*PDGFRA* (COSM22413), *n* (%)	2 (5.7%)	2 (5.7%)	1.000	1.00 (1.13–7.53)
Mutations found in only one group of patients				
*SMAD4*, *n* (%)	1 (2.9%)	0 (0.0%)	1.000	0.32 (0.01–8.23)
*STK11*, *n* (%)	1 (2.9%)	0 (0.0%)	1.000	0.32 (0.01–8.23)
*FBXW7*, *n* (%)	0 (0.0%)	1 (2.9%)	1.000	3.09 (0.12–78.41)
*JAK3*, *n* (%)	0 (0.0%)	1 (2.9%)	1.000	3.09 (0.12–78.41)
*KIT*, *n* (%)	0 (0.0%)	4 (11.4%)	0.114	10.14 (0.53–195.91)
*NRAS*, *n* (%)	0 (0.0%)	2 (5.7%)	1.000	5.30 (0.25–114.47)
*PIK3CA*, *n* (%)	0 (0.0%)	1 (2.9%)	1.000	3.09 (0.12–78.41)
*SMARCB1*, *n* (%)	0 (0.0%)	1 (2.9%)	1.000	3.09 (0.12–78.41)
***TP53***, ***n*****(%)**	**0 (0.0%)**	**10 (28.6%)**	**0.001**	**29.24 (1.64–522.00)**
*COSMIC* mutations found in only one group of patients				
*IDH1* (COSM105), *n* (%)	1 (2.9%)	0 (0.0%)	1.000	0.32 (0.01–8.23)
*JAK3* (COSM34213), *n* (%)	0 (0.0%)	1 (2.9%)	1.000	3.09 (0.12–78.41)
*KIT* (COSM21983), *n* (%)	0 (0.0%)	1 (2.9%)	1.000	3.09 (0.12–78.41)
*KIT* (COSM28026), *n* (%)	0 (0.0%)	1 (2.9%)	1.000	3.09 (0.12–78.41)
*MET* (COSM710), *n* (%)	0 (0.0%)	1 (2.9%)	1.000	3.09 (0.12–78.41)
*NRAS* (COSM584), *n* (%)	0 (0.0%)	1 (2.9%)	1.000	3.09 (0.12–78.41)
*SMARCB1* (COSM1090), *n* (%)	0 (0.0%)	1 (2.9%)	1.000	3.09 (0.12–78.41)

Bold elements denote statistical significance (*p*-value < 0.05) for pairwise comparisons between FTC and FTA. *p*-values were based on chi-square test (or Fisher’s exact test when appropriate).

**Table 2 ijms-20-03126-t002:** The coefficients necessary to determine the predicted risk of follicular thyroid carcinoma (FTC) and follicular thyroid adenoma (FTA) on the basis of single mutation *TP53*-Model 1 and the group of the mutations (comprising *FLT3* + *TP53* + *RET*) selected in the backward stepwise logistic regression-Model 2.

Mutations	*p*-Value *	β Coefficient *
Model 1		
*intercept*	0.206	−0.33
*TP53*	0.028	3.38
Model 2		
*intercept*	0.095	−0.57
*FLT3*	0.023	2.19
*RET*	0.058	−1.93
*TP53*	0.029	3.34

* Penalized maximum likelihood estimation method for bias correction (Firth’s method).

**Table 3 ijms-20-03126-t003:** The comparison of two models of FTC and FTA prediction.

Measures of Prediction Accuracy	Mutations in Logistic Regression Models		Models Comparison
	Model 1 (*TP53* Only)	Model 2 (*FLT3* + *TP53* + *RET*)	*p*-Value
AUC (95%CI)	0.64 (0.57–0.72)	0.76 (0.65–0.86)	0.016
IDI (95%CI)	0.11 (0.04–0.19)		0.003
categorial NRI (95%CI)	0.23 (0.06–0.40)		0.007
categorial NRI [FTC]	0.171		
categorial NRI [FTA]	0.057		

**Table 4 ijms-20-03126-t004:** The characteristics of patients diagnosed with follicular thyroid cancer according to their mutational status.

Characteristics	*TP53+* or *TP53-, FLT3+, RET- n* = 16	Other Mutational Status *n* = 19	*p*-Value
Male/female, *n* (%)	2/14 (12.5%/87.5%)	2/17 (10.5%/89.5%)	1.0000
Median age at diagnosis, years (range)	50 (27–81)	56 (27–82)	0.4560
Age group (≤60 years/>60 years), *n* (%)	3/13 (18.75%/81.25%)	7/12 (36.8%/63.2%)	0.2853
Median length of follow-up, months (range)	108 (17–130)	116 (28–144)	0.5674
Multifocality, *n* (%)	2 (12.5%)	2 (10.5%)	1.0000
Capsule invasion, *n* (%)	5 (31)	9 (47)	0.3322
Extracapsular extension, *n* (%)	9 (56.3%)	9 (47.4%)	0.6005
Nodal (N) involvement, *n* (%)	1 (6.3%)	3 (15.8%)	0.6081
Mean tumor size, mm (range)	32 (7–75)	29 (12–18)	0.6737
Tumor diameter ≤10 mm, *n* (%)	1 (6.3%)	1 (5.3%)	1.0000
Localization in the right/left/both lobes, *n* (%)	7/2/7 (43.8%/12.5%/43.8%)	10/1/8 (52.6%/5.3%/42.1%)	0.8844
Chronic lymphocytic thyroiditis *n* (%)	2 (12.5%)	5 (26.3%)	0.4150
Radioactive iodine-refractoriness *n* (%)	0 (0.0%)	2 (10.5%)	0.4891

The *p*-values were based on chi-square test (or Fisher’s exact test when appropriate) for categorical variables and Mann–Whitney U test for quantitative variables.

**Table 5 ijms-20-03126-t005:** Patients’ characteristics according to histopathological diagnosis.

Characteristics	Follicular Thyroid Adenomas *n* = 35	Follicular Thyroid Carcinomas *n* = 35	*p*-Value
Male/female, *n* (%)	27/8 (77.1%/22.9%)	31/4 (88.6%/11.4%)	0.205
Median age at diagnosis, years (range)	55 (29–81)	52 (27–82)	0.549
Age group (≤60 years/>60 years), *n* (%)	24/11 (68.6%/31.4%)	23/12 (65.7%/34.3%)	0.799
Median length of follow-up, months (range)	89 (18–122)	112 (17–144)	0.687
Multifocality, *n* (%)	0 (0.0%)	4 (11.4%)	0.114
Capsule invasion, *n* (%)	NA	14 (40.0%)	NA
Extracapsular extension, *n* (%)	NA	18 (51.4%)	NA
Nodal (N) involvement, *n* (%)	NA	3 (8.6%)	NA
Mean tumor size, mm (range)	23 (6–50)	30 (7–80)	0.112
Tumor diameter ≤10 mm, *n* (%)	5 (14.3%)	2 (5.7%)	0.428
Localization in the right/left/both lobes, *n* (%)	18/16/1 (51.43%/45.7%/2.9%)	17/15/3 (48.6%/42.9%/8.6%)	0.739
Chronic lymphocytic thyroiditis *n* (%)	2 (5.7%)	7 (20.0%)	0.151
Radioactive iodine-refractoriness *n* (%)	NA	2 (5.7%)	NA

The *p*-values were based on chi-square test (or Fisher’s exact test when appropriate) for categorical variables and Mann–Whitney *U* test for quantitative variables. NA—not applicable.

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
