# Peer review of "Differences in Mutational Profile between Follicular Thyroid Carcinoma and Follicular Thyroid Adenoma Identified Using Next Generation Sequencing"

_ijms, 2019, doi:10.3390/ijms20133126_

Round 1

Reviewer 1 Report

The goal of this paper is to identify genetic alterations associated with Follicular Thyroid Adenoma (FTA) and Follicular Thyroid Carcinoma (FTC) to try to obtain a novel presurgical diagnostic tool to discriminate between these two diseases. In addition, the availability of these data could help researchers to discover the molecular mechanisms underlying the tumor progression in thyroid follicular cells.

In the presented paper, Borowczyk et al. analyze the mutational status of several patients affected either by FTA or FTC. Using NGS and bioinformatic tools, the authors describe the genetic background of the thyroid lesions surgically obtained from patients.

Both mutations detection and statistical analysis appear to be correctly executed.

However, in my opinion, the results do not seem to lead to conclusions that fully reach the Author’s targets.

Major point

The clinical features of FTC are extremely heterogenous. Indeed, the group of patients includes subjects carrying tumors with or without nodal involvement; with or without radioactive iodine-refractoriness; with or without lymphocytic infiltration.

In addition, the number of analyzed subjects is small.

A segmentation of FTC subjects and a multivariate analysis could help to obtain more clear-cut conclusions.

Minor point

The discussion section is too long compared to data presented

Author Response

Dr. Martyna Borowczyk, MD, PhD

Department of Endocrinology, Metabolism and Internal Medicine

Poznan University of Medical Sciences

49 Przybyszewskiego St, 60-355 Poznan, Poland

[email protected]

June 17th, 2019

Re: Resubmission: ijms-524200

Dear Editor

International Journal of Molecular Sciences,

Many thanks for the time and effort which you and the reviewers took to review the above-referenced manuscript. We greatly appreciate the helpful suggestions for improving our manuscript and have adapted the paper accordingly as also detailed below in our point-by-point response. As a result of this revision, we believe that our manuscript has been greatly improved. Therefore, we would be grateful if you consider re-evaluating our manuscript in its revised form for potential publication in the journal.

A summary of the changes we made (detailed responses are given following this letter):

-          We have included the discussion about the data interpretation in a light of statistical calculations and a group heterogeneity.

-          We have provided more details in regard to the heterogeneity of the clinicopathological features of the patients included in the study and have added Table 3 to describe it, we have added extra calculations and underlined the results of multivariate analyses.

-          We have shortened the paper by cutting 201 words in the discussion section.

-          We have incorporated all other suggestions of the reviewer into the revised version of our manuscript.

We ensure that any revisions made in response to reviewer comments regarding patient-level data are compliant with privacy and data protection laws. All authors have read the revised manuscript and agreed to publish it in the present form.

Yours sincerely,

Dr. Martyna Borowczyk

Reviewer #1:

Comment: The goal of this paper is to identify genetic alterations associated with Follicular Thyroid Adenoma (FTA) and Follicular Thyroid Carcinoma (FTC) to try to obtain a novel presurgical diagnostic tool to discriminate between these two diseases. In addition, the availability of these data could help researchers to discover the molecular mechanisms underlying the tumor progression in thyroid follicular cells.

In the presented paper, Borowczyk et al. analyze the mutational status of several patients affected either by FTA or FTC. Using NGS and bioinformatic tools, the authors describe the genetic background of the thyroid lesions surgically obtained from patients.

Both mutations detection and statistical analysis appear to be correctly executed.

However, in my opinion, the results do not seem to lead to conclusions that fully reach the Author’s targets.

Major point: The clinical features of FTC are extremely heterogenous. Indeed, the group of patients includes subjects carrying tumors with or without nodal involvement; with or without radioactive iodine-refractoriness; with or without lymphocytic infiltration.

In addition, the number of analyzed subjects is small.

A segmentation of FTC subjects and a multivariate analysis could help to obtain more clear-cut conclusions.

Response: We would like to thank the reviewer for the opportunity to discuss the issue of conclusions’ justification and heterogeneity of the group. However, patients’ characteristics do not significantly differ, what is described in Table 2, even concerning the group’s heterogeneity. Any of the described factors, apart from the mutational status do not differ when comparing FTA and FTC – what makes the mutational status to be the only factor differentiating FTA and FTC in the multivariate regression analysis. We would like to thank the reviewer for a suggestion of FTC subjects’ segmentation and multivariate analysis to enable more clear-cut conclusions. When compared different features of FTC, what is now shown in added Table 3, we show no significant clinico-pathological differences in a group of patients divided according to their mutational status. Again, the mutational status is the only factor to differ between groups and to be included in multivariate analysis. We agree with the small subjects’ number. However, we believe that these conclusions may be confirmed on bigger data. As suggested also by reviewer 2, to the discussion section, we also have added the discussion of statistical power of the performed tests and possible generalizability of the findings.

Comment: Minor point: The discussion section is too long compared to data presented

Response: The discussion section has been revised and shortened by 201 words.

Reviewer #2:

Comment: The manuscript titled “Differences in mutational profile between follicular thyroid carcinoma and follicular thyroid adenoma identified using next-generation sequencing.” The paper provides debatable and not convincing evidence for differentiation of FTA and FTC. The manuscript is well written; provide significant insight into the background and difficulties of the distinguishing between FTA and FTC, Authors used next-generation sequencing to search for a difference in the genetic makeup of FTA and FTC. Valid hypothesis and high throughput technology to use to prove the hypothesis, result in the form of tables are a highly informative and illustrative, comprehensive discussion with relevant references.

The main criticism of this exciting and well-written manuscript is the data analysis which correctly done. However, the interpretation of the data is not convincing, and the authors should put forward a strong and irrefutable argument to support the manuscript conclusion.

 Looking at table 1, the P value of 0.030 is significant but what does it tell, in, the reality it is the Odds ratio and confidence intervals that will show the reliability of the data and subsequent validity of it. Looking at a high Odds ratio of 3.27 with a wide CI (1.22 – 8.75) shows a degree of uncertainty in accepting these data in face value.

Response: Thank you for the very kind statement concerning our paper and for suggestions to be implemented in our paper. We recognize that our study has several limitations coming from the nature of the collected data utilized in our study and the number of participants. The P value of 0.030 from Table 1 is below the statistical significance threshold, which indicates that the obtained result indicating the role of the mutational status to determine follicular lesion molecular status is not an accidental result. The statistical test dedicated to this type of analysis had small power of 0.67 here. By that, the P value was close to the threshold of statistical significance, and the obtained confidence interval for OR was wider than we would expect with such a large difference in the number of mutations. Such a result is a consequence of the insufficiently large group. For example, if we had a size group of 124 people and observed the same differences in the percentage of mutations, then the test power would increase to 0.9, and the obtained p value and the confidence interval would be significantly improved. Further studies are needed to support the study conclusions. As suggested by reviewer 1, we have added Table 3 and multivariate analysis to better justify the study’s conclusions. We have added an appropriate paragraph to the discussion section. We would like to continue a project in further studies.

Comment: For TP53, a P value of 0.001 and Odds ratio of 29.24 with CI of (1.64 - 522) shows the data are not showing reliability. Authors should provide a convincing argument to satisfy the scientific community why they should believe that these data are correctly interpreted.

 The manuscript has merit and is of interest to scientific and medical communities.

Response: We would like to thank for the kind words about merit and possible interest of the paper. In regard to statistical doubts, for TP53, a P value was 0.001 and it was well below the statistical significance threshold. This is strong evidence of the existence of a real TP53 relationship with the occurrence of the FTC. Indeed, there are situations when the very wide confidence interval for the OR indicates the uncertainty of the result. However, this situation does not belong to them. The whole range is above 1, and increasing its width only increases the uncertainty as to how much higher the chance for FTA when occurring in the TP53 mutation, and not whether this chance is greater. Please allow us to illustrate two examples:

1) In our case:

• none (0% of the 35 people) with FTA had this mutation

• 29% (10 out of 35 people) from the FTC group had this mutation,

which confirms that more often mutations appear in the FTC group. We see a wide confidence interval, i.e. the chance for an FTC is from almost two to as high as 522 times higher when there is a TP53 mutation.

2) If we had a situation in which:

• none (0% of the 35 people) with FTA had this mutation

• 97% (34 out of 35 people) from the FTC group had this mutation,

we would be even more sure about the TP53 relationship with the occurrence of mutations, and we would still get a very wide confidence interval - i.e. from 64 to 41477. However, this range, although still wider all the time would be statistically significant (p <0.001), which would only increase the certainty that TP53 is an unmistakable FTC prediction marker.

The wide confidence interval (such a distal upper limit of this range) is a consequence of the lack of people with the TP53 mutation in the FTA group. The wide confidence interval above 1 increases the uncertainty of how much the TP53 mutation increases the FTC risk and not whether the TP53 mutation increases the FTC risk.

We agree, however, that the wide confidence interval for the OR may raise doubts. The work also uses other methods of analysis, including logistic regression model and the area under the ROC curve, which also confirms the statistical significance of the FTC compound with the TP53 mutation. That's why we'll put the following explanation in the paper:

The wide confidence interval for the odds ratio does not allow to precisely determine how many times higher the risk for FTC diagnosis will be when this mutation occurs, but it does not undermine the statistical significance of the relationship. The significant influence of the TP53 mutation on the occurrence of FTC was also confirmed by building a logistic regression model and determining the ROC curve (Table 3b), where we can also see that the area under the ROC curve with the confidence interval (AUC (95% CI) = 0.64 (0.57-0.72)) is above 0.5, which increases the reliability of the findings.

Reviewer 2 Report

The manuscript titled “Differences in mutational profile between follicular thyroid carcinoma and follicular thyroid adenoma identified using next-generation sequencing.” The paper provides debatable and not convincing evidence for differentiation of FTA and FTC. The manuscript is well written; provide significant insight into the background and difficulties of the distinguishing between FTA and FTC, Authors used next-generation sequencing to search for a difference in the genetic makeup of FTA and FTC. Valid hypothesis and high throughput technology to use to prove the hypothesis, result in the form of tables are a highly informative and illustrative, comprehensive discussion with relevant references.

The main criticism of this exciting and well-written manuscript is the data analysis which correctly done. However, the interpretation of the data is not convincing, and the authors should put forward a strong and irrefutable argument to support the manuscript conclusion.

 Looking at table 1, the P value of 0.030 is significant but what does it tell, in, the reality it is the Odds ratio and confidence intervals that will show the reliability of the data and subsequent validity of it. Looking at a high Odds ratio of 3.27 with a wide CI (1.22 – 8.75) shows a degree of uncertainty in accepting these data in face value. For TP53, a P value of 0.001 and Odds ratio of 29.24 with CI of (1.64 - 522) shows the data are not showing reliability. Authors should provide a convincing argument to satisfy the scientific community why they should believe that these data are correctly interpreted. 

 The manuscript has merit and is of interest to scientific and medical communities.

Author Response

(The authors gave the same response as above.)

Round 2

Reviewer 1 Report

THE AUTHORS HAD ADDRESSED  REVIEWER'S CONCERNS